# Bat Rabies in the Americas: Is *Myotis* the Main Ancestral Spreader?

**DOI:** 10.3390/v16081302

**Published:** 2024-08-16

**Authors:** Diego A. Caraballo, María Lorena Vico, María Guadalupe Piccirilli, Stella Maris Hirmas Riade, Susana Russo, Gustavo Martínez, Fernando J. Beltrán, Daniel M. Cisterna

**Affiliations:** 1Instituto de Ecología, Genética y Evolución de Buenos Aires, CONICET, Facultad de Ciencias Exactas y Naturales, Universidad de Buenos Aires, Ciudad Autónoma de Buenos Aires C1428EHA, Argentina; 2Departamento de Zoonosis Urbanas, Ministerio de Salud de la Provincia de Buenos Aires, Buenos Aires B1870, Argentina; 3Servicio de Neurovirosis, Instituto Nacional de Enfermedades Infecciosas, Administración Nacional de Laboratorios e Institutos de Salud (ANLIS), “Dr. Carlos G. Malbrán”, Ciudad Autónoma de Buenos Aires C1282AFF, Argentina; 4Dirección General de Laboratorio y Control Técnico (DILAB), Servicio Nacional de Sanidad y Calidad Agroalimentaria (SENASA), Buenos Aires B1640CZT, Argentina; 5Instituto de Zoonosis “Luis Pasteur”, Ciudad Autónoma de Buenos Aires C1405DCD, Argentina

**Keywords:** bats, rabies, phylogeny, cross-species transmission, host shift, spillover

## Abstract

The rabies virus (RABV) is the exclusive lyssavirus affecting both wild and domestic mammalian hosts in the Americas, including humans. Additionally, the Americas stand out as the sole region where bat rabies occurs. While carnivore rabies is being increasingly managed across the region, bats are emerging as significant reservoirs of RABV infection for humans and domestic animals. Knowledge of the bat species maintaining rabies and comprehending cross-species transmission (CST) and host shift processes are pivotal for directing surveillance as well as ecological research involving wildlife reservoir hosts. Prior research indicates that bat RABV CST is influenced by host genetic similarity and geographic overlap, reflecting host adaptation. In this study, we compiled and analyzed a comprehensive nucleoprotein gene dataset representing bat-borne RABV diversity in Argentina and the broader Americas using Bayesian phylogenetics. We examined the association between host genus and geography, finding both factors shaping the global phylogenetic structure. Utilizing a phylogeographic approach, we inferred CST and identified key bat hosts driving transmission. Consistent with CST determinants, we observed monophyletic/paraphyletic clustering of most bat genera in the RABV phylogeny, with stronger CST evidence between host genera of the same family. We further discuss *Myotis* as a potential ancestral spreader of much of RABV diversity.

## 1. Introduction

Rabies (RABV) is the most widespread and oldest known lyssavirus infecting mammals, both domestic and wild, including humans. RABV (genus: Lyssavirus, family: Rhabdoviridae) causes acute, highly lethal encephalomyelitis and is responsible for numerous local epidemics, resulting in approximately 59,000 human deaths annually, primarily in Africa and Asia [1]. While Rhabdoviruses can infect a very wide range of hosts, including plants, invertebrates, and vertebrates via vectors, as well as vertebrates without vectors, the lyssaviruses are unique in that they are the only group known to exclusively infect mammals, without exception [2]. Bats are the primary hosts of most lyssaviruses, but rabies is unique in that it circulates not only among mammals of the order Chiroptera but also among a diverse set of Carnivora [3]. Occasional spillover infections occur in other animal species, including humans and domestic animals.

Phylogenetically, RABV can be divided into two primary groups: (i) one comprising viruses sampled from bats and carnivores in the Americas, referred to as the ‘bat-related’ virus group, and (ii) another primarily consisting of ‘dog-related’ viruses sampled from carnivores, excluding bats [4]. The ‘bat-related’ RABV group is limited to viruses circulating among various New World bat species, as well as some mesocarnivores such as skunks and raccoons, forming a distinct phylogenetic cluster within this group [3,4]. In Latin America, significant attention has been directed towards vampire bats as hosts, particularly due to their role as reservoir species for RABV, capable of infecting other mammals, including humans and livestock [5,6]. Nevertheless, there is a significant occurrence of spillovers, including the establishment of variants, from insectivorous bats to other mammals. This has resulted in the conceivable establishment of RABV variants in various species, not only in carnivores such as skunks, raccoons, foxes, coatis, and kinkajous but also in marmosets and presumably in capuchin monkeys [7,8,9,10]. These spillover events also carry significant public health implications, as demonstrated by a case where the virus was transmitted from *Tadarida brasiliensis* to a cat, resulting in a fatal human infection in Argentina through secondary transmission in 2021 [11].

The main determinants of bat RABV cross-species transmissions (CST) have been shown to be host genetic similarity and, to a lesser extent, geographic overlap between species from different genera [12,13]. In these studies, based on North American bat RABV, higher host genetic similarity correlated with a notably increased occurrence of both spillovers (in external nodes) and host shifts (limited to internal nodes). This is attributable to the assumption that highly related hosts will require less viral adaptation due to finding a similar physiological and immunological environment [14].

An expectable outcome of host adaptation is that virus occurrence will be constrained by host phylogeny. This was demonstrated by the non-random clustering of 34 RABV host species in the carnivore phylogeny [15]. This would imply that, in a phylogenetic reconstruction of RABV variants, host taxa will form monophyletic/paraphyletic clusters.

In this study, we generated a dataset representing the diversity of bat-borne RABV in Argentina to contextualize it within the broader spectrum of bat RABV diversity in the Americas through a Bayesian phylogenetic approach. We further analyze the association between host genus and geography, revealing that both factors contribute to the global phylogenetic structure. Employing a phylogeographic approach, we aimed to model CST and determine which bat hosts act as main spreaders. Based on the background on CST determinants and adaptation, we predict that (i) most bat genera will be monophyletic/paraphyletic in the RABV phylogeny, indicative of viral adaptation and (ii) evidence for CST will be higher between host genera of the same family. We further analyze the role of *Myotis* as the potential ancestral spreader of the majority of RABV diversity observed at the present.

## 2. Methods

### 2.1. Samples

A total of 62 brain samples from RABV-positive insectivorous bats were obtained from 2001 to 2023 from several Argentinian provinces (Figure 1, Appendix A): Buenos Aires (50), Chubut (1), Córdoba (7), La Pampa (1), Río Negro (2), and Tierra del Fuego (1). These samples belong to the following bat species: *Tadarida brasiliensis* (50), *Myotis dinelli* (3), *Myotis levis* (3), *Myotis* sp. (1), *Eptesicus furinalis* (2), *Dasypterus ega* (1), *Aeorestes cinereus* (1), and *Eumops bonariensis* (1). These specimens were gathered as part of rabies virus surveillance conducted by the Avellaneda Zoonosis Center in Buenos Aires province (BAI), the Luis Pasteur Zoonosis Institute in the Autonomous City of Buenos Aires (CABA), and the National Agri-Food Health and Quality Service (SENASA) in Buenos Aires province.

### 2.2. RNA Extraction, PCR Amplification, and Sequencing

Viral RNA was obtained using the QIAmp Viral RNA Mini kit, Qiagen, following the manufacturer’s instructions. Reverse transcription and PCR amplification were performed using random primers for retrotranscription and primers Lys001, 550B, JW12, 550F, and 304R to obtain the complete RABV nucleoprotein gene, using PCR conditions as described in [16]. The product obtained was purified using the ExoSAP-IT reagent, Usb; Nucleotide sequences were obtained using the BigDye Terminator 3.1 reagent (Applied Biosystems, Waltham, MA, USA), purified with the XTerminator reagent (Applied Biosystems), and sequenced using the ABI 3500 genetic analyzer (Applied Biosystems). The sequences generated were deposited in GenBank under the accession numbers PP661584-PP661645 (Appendix A).

### 2.3. Identification of Bat RABV Main Lineages

An initial analysis was conducted to identify all major bat-borne RABV (rabies virus) lineages. Subsequently, a subsample was selected to encompass all primary host lineages and sublineages, as well as the countries where these lineages were sampled. This set the stage for a secondary analysis aimed at testing the phylogenetic positioning of Argentinian samples and conducting host-switching analysis.

Reference sequences for the initial phylogenetic analysis were retrieved from the NCBI virus database (https://www.ncbi.nlm.nih.gov/labs/virus/vssi/, accessed on 1 April 2023) [17], using the following filters: Virus: Lyssavirus rabies (taxid: 11292), Host: Chiroptera (bats) (taxid: 9397), Protein: Nucleoprotein, and filtered for samples from the Americas. The search yielded 1864 hits. These sequences were aligned with Clustal Omega and subjected to an initial analysis on IQ-TREE [18], using the GTR + F + R5 model selected with the corrected Akaike Information Criterion (AICc). This preliminary analysis enabled the distinction of 22 RABV main host lineages (Appendix A). A subset of complete nucleoprotein sequences that represent all main lineages/sublineages (main host species) and all countries was selected for testing the sequences obtained in this study (Appendix A). The validity of host species and genus names was verified using the Integrated Taxonomic Information System (ITIS) online database, www.itis.gov, CC0 (https://doi.org/10.5066/F7KH0KBK, accessed on 1 February 2024) and updated to reflect current taxonomy when necessary.

### 2.4. Bayesian Phylogeny and Cross-Species Transmission Dynamics

The complete matrix consisted of 132 sequences (62 generated in this study plus 70 N sequences from bats representing all main lineages, see Appendix A). A Bayesian phylogeny was run using BEAST 1.10.4 [19]. The molecular substitution model was selected with MrModeltest v2 using the second-order Akaike information criterion (AICc) [20], separating each codon position (GTR + I + G, HKY + I + G, and GTR + G, for 1st, 2nd, and 3rd codon positions, respectively). A genus-level host-switching dynamics analysis was performed using an asymmetric model, allowing for unequal transition rates. This method enables the reconstruction of ancestral host posterior probabilities at each internal node of the tree and estimates the transition rates between states (hosts). A tip-dated strict molecular clock was utilized, alongside a strict clock for the host genus. A Bayesian stochastic search variable selection (BSSVS) method was applied to assess the significance of transitions between trait states, using Bayes factors (BFs) as a measure of statistical significance. BFs were computed using SpreaD3 [21], and their interpretation followed Jeffreys’ guidelines [22]: (BF > 3: substantial support, BF > 10: strong support, BF > 30: very strong support, BF > 100: decisive support). Only transitions strongly supported by BFs are shown in figures, consistent with methodologies employed in prior studies. A coalescent with a constant-size tree prior was selected. A total of 1E8 Markov chain Monte Carlo (MCMC) generations were run, sampling every 1E4 generations, discarding the first 10% of the run (burn-in phase). Convergence of the MCMC runs was confirmed using Tracer. Trees were visualized in FigTree v1.4.4 [23] and plotted with ggtree v3.6.2 [24] and ggplot2 version 3.4.2 [25], using R version 4.2.3 [26] and RStudio version 2023.03.0 Build 386 [27].

The decision to base our analysis at the genus level has multiple reasons. Primarily, it allows us to reflect profound host shifting events, particularly those occurring between genera. This approach allows for a comprehensive analysis of the contribution of each bat genus, rather than focusing solely on species. Analyzing at the species level may lead to inflated rates of CST by potentially overlooking common variants shared among closely related species. This approach could also diminish the significance of a donor bat genus’s contribution to a recipient one. Additionally, our choice is grounded in the frequent challenges associated with species-level identification in bats. The prevalence of cryptic species and the potential for human error in identification underscore the practicality and reliability of analyzing at the genus level [28].

### 2.5. Association Tests

Statistical tests were conducted to examine trait clustering among the tips of the phylogeny. Bayesian Tip Association Significance Testing (BaTS version 0.9 beta) software was utilized for this purpose [29]. We assessed the clustering patterns of the bat genera and countries sampled. To evaluate the significance of trait clustering, we compared the association index (AI), parsimony score (PS), and monophyletic clade (MC) statistics derived from posterior samples of each run (with a burn-in of 10%) against null distributions created from 100 randomizations of traits assigned to tips across the sampled trees. A significance level of *p* < 0.05 was chosen for this analysis.

### 2.6. Recombination Analysis

To identify candidate recombinants, the alignment including 132 bat-related nucleoprotein sequences was tested for recombination using the Recombination Detection Program RDP version 4 (RDP4) [30]. Default settings for all algorithms were applied (RDP, GENECONV, Maxchi, Chimaera, BootScan, SiSscan, and 3Seq), incorporating SiSscan “primary scans” (the default option is that this method is only used to examine sequences in which recombination signals are detectable by other “primary scanning” methods). For each inferred event, a phylogenetic analysis was run in IQ-TREE using the recombinant segments to test for topological incongruence, estimating node support using an approximate likelihood-ratio test (1000 replicates).

An additional analysis to detect recombination was conducted using the GARD method [31], on the Datamonkey 2.0 web server [32]. The analysis employed Beta-Gamma site-to-site rate variation and utilized four classes to model rate heterogeneity.

### 2.7. Genetic Distances

Uncorrected pairwise genetic p-distances were calculated with MEGA X [33]. Distances were calculated by considering all sequences belonging to a specific lineage, defined as members of either a monophyletic or paraphyletic group within a given host genus. Some genera comprised multiple lineages, each treated separately. Furthermore, distances between all sequences from all lineages within the genera *Eptesicus*, *Myotis*, and *Tadarida* were calculated to evaluate overall variation within these widely distributed genera represented by multiple lineages in the tree. In the distance calculation of *Myotis yumanensis*, two sequences were excluded (KY203184 and KY203142), since these were considered in the calculations of the lineage “*Myotis* NA” (see below). Similarly, one sequence of *Myotis austroriparius* (AY039225) was excluded from the intralineage distance, because it pertains to a non-related lineage with respect to the other two sequences.

## 3. Results

### 3.1. Recombination Analysis

Detecting recombinants prior to phylogenetic analysis ensures the accuracy of evolutionary reconstructions by preventing the inclusion of sequences that may introduce conflicting signals and distort inferred relationships. Three putative recombination events were detected with RDP4. Event 1 is supported by four out of seven methods and involves two sequences from *Tadarida* (major parent and recombinant) and one from *Eumops* in the *Aeorestes* clade. The recombinant fragment spans nucleotides 718 to 828 (111 bp in length). It appears to be a true recombinant because the sequence is identical to the minor parent in the putative recombinant region (15 substitutions in 111 sites), as confirmed by a quick phylogenetic test conducted with IQ-TREE. However, this sequence was not generated in our analysis (GenBank accession number: JF826130), and thus, we cannot confirm if it was produced in error (contamination, assembly error, etc.). Even if it is a true recombinant, the involvement of such a short fragment causes this sequence to remain in the parental clade when the complete alignment is included. The second event involves two sequences from the *Myotis* SA clade (major parent and recombinant) and a sequence from *Tadarida* SA. The breakpoints of this event are not unambiguously identified, and in the recombinant fragment, there is no clear affinity to the minor parent (it shares multiple interspersed substitutions, comparable in number, with both parental sequences). This potential event is more likely to represent a divergent sequence rather than a recombinant. The third event lacks a clear signal of recombination and was detected only by one method (SiScan); thus, it is not considered a true recombination event.

In addition, GARD found no evidence of recombination. A total of 570 potential breakpoints were identified, of which only one was confirmed as a breakpoint. However, in the topological analysis, the multiple tree model could not be preferred over the single tree model by an evidence ratio of 100 or greater, indicating the inferred breakpoint may reflect rate variation instead of topological incongruence.

Taken together, the lack of evidence of recombination in the dataset allowed us to proceed with the phylogenetic inference and downstream analyses.

### 3.2. Molecular Phylogeny

A total of 62 N gene whole sequences were generated in this study. No premature stop codons were found in the dataset. Although there is low support near the base of the tree, there are four well-supported groups that encompass the diversity of bat RABV (Figure 2). Appendix A contains the annotated tree, allowing exploration of additional data such as sample country annotations and support values for internal nodes. It is interesting to note that except for *Myotis*, *Eptesicus*, and *Dasypterus*, all bat genera are lineage-specific, occurring in a single monophyletic or paraphyletic cluster in the tree. The most-represented lineage is that of *Tadarida brasiliensis* from South America (*Tadarida* SA), which comprises 54 closely related sequences, including samples from Brazil, Argentina, Chile, and Uruguay. As confirmed in the preliminary analysis (Appendix A), *Tadarida* is paraphyletic with respect to *Desmodus*/*Artibeus*, being the North American variant (*Tadarida* NA) sister to this group. Similarly, the *Lasiurus*/*Dasypterus*/*Aeorestes* clade is paraphyletic to a clade containing variants circulating in *Perimyotis* and *Lasionycteris*. It is noteworthy that there are sequences from both subcontinents in this clade, suggesting that these bats are connected through migration. However, it is remarkable that the structure of this clade, comprising three closely related genera, is more influenced by geographical factors (country) than by the host species (Appendix A). Another well-supported clade is that of *Histiotus* (and *Eptesicus*) from Brazil and Argentina, which is paraphyletic to a variant found in *Nyctinomops*. This clade is related to four lineages found in *Corynorhinus townsendii*, *Myotis volans*, *M. yumanensis*, and *Eptesicus fuscus*, from the USA and Canada. Closely related *Eptesicus* and *Myotis* variants are also found in Brazil and Argentina, including RABV sequences obtained from *Myotis levis* and *Myotis dinelli*, and those obtained from *E. furinalis*. There is an additional *Myotis* clade from Canada and the USA which is paraphyletic with respect to *Parastrellus* and *Antrozous*. Finally, there is a third *Eptesicus* clade from the USA (*Eptesicus* NA2).

### 3.3. Association Tests

In the phylogeny, there is a clear hierarchical structure that responds to two levels: host and geography. Both global tests, PS and AI, indicate that the association exists at both levels. This suggests that related RABV sequences are more likely to share a host or geographic location by ancestry than they do by chance. At the host level, the genera that contribute to stronger host–virus associations (producing larger monophyletic clades) are *Tadarida* and *Myotis*, although there are several other genera with significant associations (Appendix A). At the geographic level, Argentina, the USA, and Canada are characterized by the strongest virus–country associations (Appendix A).

### 3.4. Cross-Species Transmission Dynamics

Two types of CST can be distinguished: those that took place in the recent past, and likely represent dead-ends in the transmission chain (spillovers), and those that can be inferred from ancestral states reconstruction, and that reflect the adaptation of the virus to a new host (Figure 2). Among the sequences generated in this study, an *Aeorestes*→*Eumops* spillover was confirmed alongside two *Myotis*→*Eptesicus* events in independent clades of the tree (Figure 2). One occurred in a clade sister to the *Histiotus* lineage, while the other took place in the clade *Myotis* SA. Remarkably, both spillovers involved *M. dinelli* and *E. furinalis*, which may be indicative of an ecological association between these species.

The ancestral host reconstruction poses *Aeorestes*/*Dasypterus* as the most probable common ancestor of bat rabies. However, it should be taken into account that the relationships between the main clades are not strongly supported, and this result should be then taken cautiously.

The Bayesian stochastic search variable selection (BSSVS) procedure was employed to identify viral host switches between bat genera along the branches of the MCC tree, further calculating Bayes factors (BF) to estimate the significance of these switches. Fourteen highly supported (BF < 10) inter-genus host switches were identified (Figure 3). The highest supported switches (BF > 100) occurred from *Myotis* to *Eptesicus*, and from *Desmodus* to *Artibeus*. *Myotis* is the most frequent donor, with significant host switches to five other genera. This approach enables the identification of historical transmission chains such as *Tadarida* → *Desmodus* → *Artibeus*, and *Myotis*/*Eptesicus* → *Histiotus* → *Nyctinomops*. In addition, host shifts in both directions were found between *Aeorestes*/*Dasypterus* and *Lasionycteris*/*Perimyotis*. This could be indicative of recent host divergence (*Aeorestes*/*Dasypterus*) or ecological association, in the case of less-related taxa.

### 3.5. Genetic Distances

The estimates of evolutionary divergence between sequences varied significantly across different lineages (Figure 4, Appendix A). Among the 22 lineages analyzed, 14 exhibited low levels of divergence (<0.025 substitutions per site). Certain lineages, such as *Desmodus*, *Eptesicus* NA2, and *Myotis* NA, displayed intermediate levels of genetic divergence. Although these values and their distribution may be influenced by the limited number of samples in some lineages, it is noteworthy that *Histiotus* and *Parastrellus*, despite their low representation in the dataset, exhibited high levels of genetic divergence. This suggests that these lineages experienced elevated substitution rates and/or reduced homogenization. In contrast, the lineage *Tadarida* SA, which possesses the largest representation in the dataset (comprising 54 sequences from four countries), displayed low levels of genetic differentiation, indicating significant homogenization among these bats.

Furthermore, lineages such as *Eptesicus* NA1 and *Myotis* SA displayed significant dispersion, where closely related sequences formed sublineages; however, these sublineages exhibited considerable divergence from each other. The highest levels of divergence, accompanied by significant dispersion, were observed in *Dasypterus*, with distance values exceeding 0.1 substitutions per site. This reflects both the lack of monophyly within this lineage and its extensive geographical distribution, spanning from South America to North America.

In the comparison of genetic distances at the genus level, the three genera analyzed showed contrasting patterns (Figure 5). *Eptesicus* has high intra-genus distances (mean 0.1 substitutions per site), reflecting the phylogenetic distance between its four non-related clades. *Eptesicus* NA2 originated from a host shift from *Myotis* and is more related to *Eptesicus* SA than to the other *Eptesicus* lineage from North America (even when *Eptesicus* NA1 and NA2 both circulate among *E. fuscus*, while *Eptesicus* SA circulates in *E. furinalis*). *Myotis* also depicts high intra-genus distances, in concordance with the phylogenetic diversity observed in RABV variants circulating in this genus. On the contrary, *Tadarida* exhibits a distinct bimodal pattern, indicative of the separation between North American and South American clades. The majority of distance values correspond to intra-clade differences (Tadarida SA), resulting in an exceptionally low mean of 0.02 substitutions per site.

## 4. Discussion

Surveying virus diversity is crucial for understanding viral ecology and developing effective strategies for disease prevention and control. In this study, we obtained and analyzed full nucleoprotein sequences of bat-related RABV in Argentina. Previous studies involved short fragments (191–320 bp) that, although allowing for variant distinction, were less resolutive at internal nodes of the phylogeny [34,35,36]. In addition, this study focuses on all available bat RABV sequences, encompassing the entirety of known virus diversity to date, and explicitly conducts a CST analysis. Research on CST has predominantly centered on lineages originating from North America, neglecting other geographical regions [12,13,37,38]. The single prior study examining the diversity across both subcontinents conducted a phylogenetic analysis utilizing partial N gene sequences (582 bp) [39]. This research was limited in its representation of several lineages, which were identified in the comprehensive phylogenetic analysis conducted in the present study. As a result, these additional lineages were included in both the phylogeny and the CST analysis. Therefore, the present study integrates more robust information (full N gene and all known bat RABV lineages) from both subcontinents, providing insights, particularly into widespread host taxa.

The phylogeny revealed a clear hierarchical structure that responds to two levels: host and geography. This result was further supported by the tip-association tests. The majority of bat genera are lineage-specific, confirming our first prediction. Interestingly, not only geographically restricted genera (e.g., *Histiotus*, *Antrozous*) but also three widespread genera found across the three American subcontinents, *Tadarida*, *Desmodus*, and *Artibeus*, represent this situation. *Tadarida* is divided into two related subclades, South America and North America, while *Desmodus* and *Artibeus* share RABV sequences that represent a unique monophyletic variant. This result was also confirmed by a phylogeographic study of vampire bat-associated RABV that revealed that the viruses found in bats from both North and South America originate from a single common ancestor, indicating that they constitute a singular, well-defined epizootic event [40].

In contrast to strict host specificity, the presence of multiple lineages within a host genus reflects dynamics dominated by host switching. We observe that most widespread host genera cluster in different non-related RABV lineages, suggesting that when species/populations are geographically isolated, there is more chance of variant extinction and recolonization by a less-adapted variant. The presence of multiple viral lineages within the widespread host genera *Eptesicus*, *Dasypterus*, and *Myotis a priori* (but see below for *Myotis*) suggests that their species acquired RABV infections independently, leading to simultaneous epizootics in different regions. *Eptesicus* is divided into three clades, two of which represent the North American *E. fuscus*, while the other corresponds to the South American *E. furinalis*. As occurs in *Myotis*, in North America, *Eptesicus* has two non-related cycles (both are related to *Myotis*). In the case of *Dasypterus*, it is interesting to note that its variants are interspersed among other closely related genera (*Aeorestes* and *Lasiurus*), and this may be indicative of ecological and phylogenetic proximity.

The presence of four distinct *Myotis* lineages in North America (Figure 2) might initially suggest multiple independent acquisitions of RABV through host shifts, as proposed in prior studies [13,37,39]. However, upon considering the ancestral host reconstruction and CST dynamics inferred from the BSSVS analysis, a different interpretation emerges. The major bat RABV clade, which encompasses all *Myotis* lineages alongside most host lineages, except for *Dasypterus*/*Lasiurus*/*Aeorestes* (plus *Lasionycteris* and *Perimyotis*), indicates *Myotis* as the common ancestor of these lineages. In the host shifting analysis (Figure 3), *Myotis* emerges as the predominant donor and never as a (significant) recipient of RABV. Taken together, these results suggest that *Myotis* did not acquire RABV variants from other bat genera; rather, its numerous species and populations harbored most of the observed RABV diversity in bats today and transferred RABV to other bat genera. The occurrence of multiple RABV lineages in *Myotis* aligns with its status as an ancient (>33 Mya) and globally distributed genus, encompassing over 120 described species [41]. Indeed, these bats host a wide variety of viruses, such as coronaviruses, astroviruses, and picornaviruses, and they have been identified as key players in the diversification of these pathogens [42,43,44]. The diversification of the genus *Myotis* resulted from a rapid speciation process [41], which was shaped by the isolation and reconnection of populations during the climatic fluctuations of the Quaternary, ultimately making it the most speciose mammal genus globally [45]. Throughout this evolutionary journey, accompanying pathogens probably co-diversified with *Myotis*, undergoing similar isolation and reconnection dynamics that elucidate the diversity of RABV observed in present times. It is likely that *Myotis* acquired RABV after the divergence of the New World and Old World clades, about 19 MYA [46] or that it was lost in the Old World lineage. Both possibilities are plausible for explaining the absence of bat RABV in the Old World.

Two contrasting patterns of genetic distances used as a proxy of phylogenetic diversity were found (Figure 4 and Figure 5). Some lineages, such as *Histiotus* and *Parastrellus*, depicted high levels of genetic divergence, suggesting that they could be experiencing high rates of substitution rates or reduced homogenization. However, these values are based on a few sequences, and a more detailed analysis should be carried out. In contrast, the lineage *Tadarida* SA, which possesses the largest representation in the dataset (comprising 54 sequences from four countries), displayed low levels of genetic differentiation, indicating significant homogenization among these bats, probably due to high levels of gene flow that characterizes this species [47].

Previous studies of North American bats across various genera revealed that the occurrence of sustained transmission and spillover of RABV increased with higher host phylogenetic relatedness [13,48], likely due to the lower number of adaptive changes necessary for viral exchange. However, between closely related species (within a genus), no significant relationship between host species relatedness and the frequency of CST events was found [37]. At this level, where all host species are highly related, ecological factors, such as the level of overlap in geographic ranges, gain preponderance. In the present study, we confirmed a predominance of host shifts among members of the same bat family, reflecting the existence of evolutionary constraints at this taxonomic level.

The two highest supported CST were *Myotis* → *Eptesicus*, and *Desmodus* → *Artibeus*. It is worth noting that both pairs of CST were unilateral, where the inferred donor hosts are *Myotis* and *Desmodus*, and there are multiple host shifts to the recipient hosts *Eptesicus* and *Artibeus*, supported also by the broader phylogenetic analysis (Appendix A). A significant association between *Myotis* and *Eptesicus* is evident, as all *Myotis* lineages show an associated *Eptesicus* lineage [49]. While previous claims suggested that *Desmodus* and *Artibeus* maintained independent RABV cycles, clustered in reciprocally monophyletic lineages [50,51], the approach employed in this study suggests a more intricate scenario. The ancestral host is likely *Desmodus* which received RABV from North American *Tadarida*. *Artibeus* might have acquired the virus via CST from *Desmodus* twice, but numerous spillbacks *Artibeus* → *Desmodus* were also observed (see Appendix A). Since we aimed to include representatives of main host lineages and not spillovers to capture host shifts, in our CST approach we included only one member of each *Artibeus* lineage (Figure 2), so we could not quantify the level of spillbacks to *Desmodus*.

The results presented in this study confirm the general paradigm that CST of viruses is more likely to occur among genetically similar taxa, suggesting that genetic relatedness of hosts may imply similar immune and physiological environments, reducing barriers to adaptation to a new host [14]. However, there is considerable evidence demonstrating that bat-related RABVs have repeatedly jumped to several terrestrial and arboreal New World mammal species, including a wide range of mesocarnivores (cats, dogs, foxes, skunks, coatis, and kinkajous) but also primates (marmosets and capuchin monkeys) [7,8,9,52]. Some of these events led to the establishing of independent transmission cycles, in which these mammalian species also became RABV reservoirs. The raccoon (*Procyon lotor*) strain of the rabies virus has a shared ancestor with a skunk strain, and both are most closely related to rabies viruses found in bats [53]. In white-nosed coatis, a variant derived from *Tadarida brasiliensis*, has also been demonstrated to represent a stable host shift [54], while in marmosets, RABV underwent a stable host shift likely from bats of the genus *Lasiurus* [55]. Even in cases where stable host shifts have not been confirmed, it is common to observe repeated host shifts from the same donor to the same recipient host species, involving the same RABV variant. This phenomenon, together with the finding of pre-shift convergent mutations led to the hypothesis that significant pre-adaptation of bat viruses should take place before effective host shifting to terrestrial mammals [9]. The factors determining the success in bat-to-carnivore spillovers and stable host shifting are far from being elucidated but may include social behavior, lifespan, reproductive cycle duration, population density, and home range, among others [56]. The potential of wild and domestic mesocarnivores to be successful RABV reservoirs, in addition to the host shifts and spillovers cited above, should be an alert for the need of rapid diagnostics and typing in RABV surveillance for effective disease prevention and mitigation. Considering that bat rabies is non eradicable, in countries where canine rabies has been completely (or almost) completely eliminated, these bat-to-mesocarnivore spillovers may be the main cause of risk of spillover to humans.

In this study, we investigated the diversity of bat RABV from Argentina, placing it in the context of the total diversity known to date. We confirmed our first hypothesis that most bat hosts are clustered in single RABV lineages. Among widespread genera, we identified two patterns: those that acquired independent RABV variants multiple times (e.g., *Eptesicus* and *Dasypterus*) and *Myotis*, proposed as the ancestral host genus of the major bat RABV clade and the primary donor to other genera. In addition, we corroborated our second prediction that the majority of CST would occur between related bat genera (same family) and underscored the strong association between *Myotis*/*Eptesicus* and *Desmodus*/*Artibeus*, supported by both host shifts and spillovers. Although the panorama is far from complete—for example, there is a large gap in molecular data from surveillance in Central America—the identification of main hosts and the characterization of CST can be useful for better understanding RABV epizootiology and preparedness for viral emergence.

## Figures and Tables

**Figure 1 viruses-16-01302-f001:**
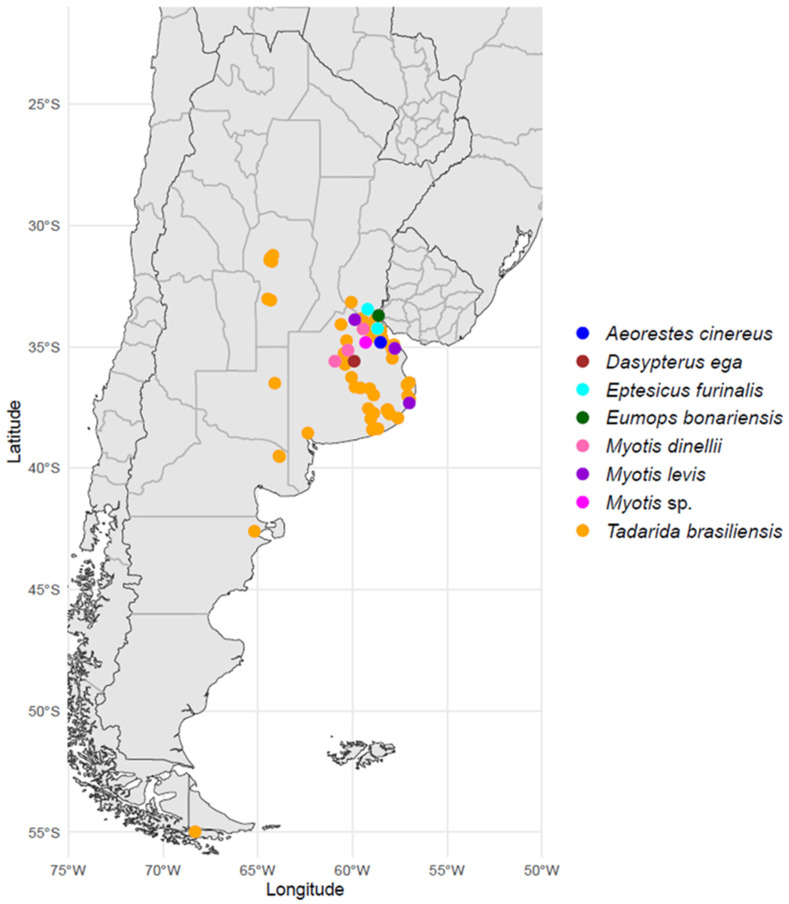
Sampling localities and bat host species of the RABV N sequences generated in this study.

**Figure 2 viruses-16-01302-f002:**
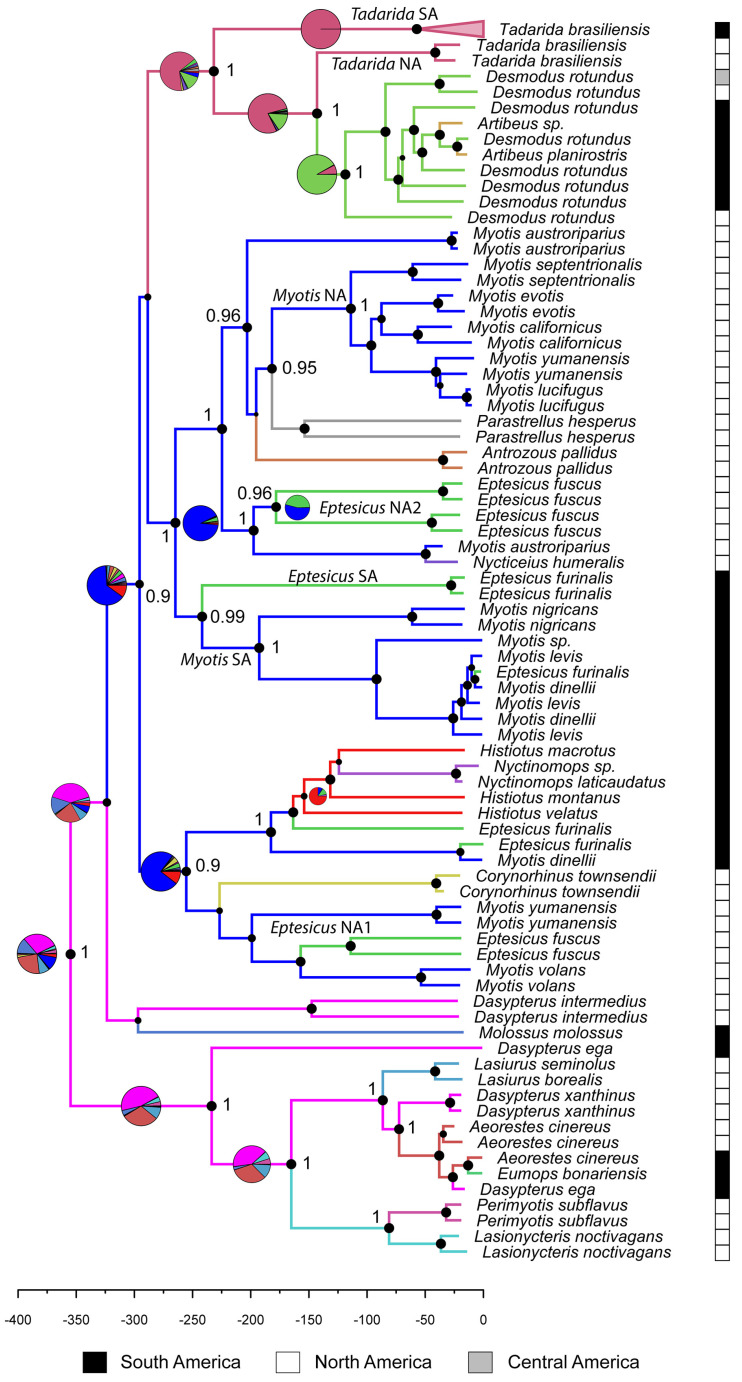
Maximum clade credibility tree of the Bayesian phylogenetic analysis of bat RABV representing the whole diversity of the virus. Node sizes are proportional to Bayesian posterior probability (relevant nodes are shown with numeric values). Different colors were assigned to each bat genus. Pie charts represent uncertainty in the ancestral host reconstruction. The timescale is in years before the present.

**Figure 3 viruses-16-01302-f003:**
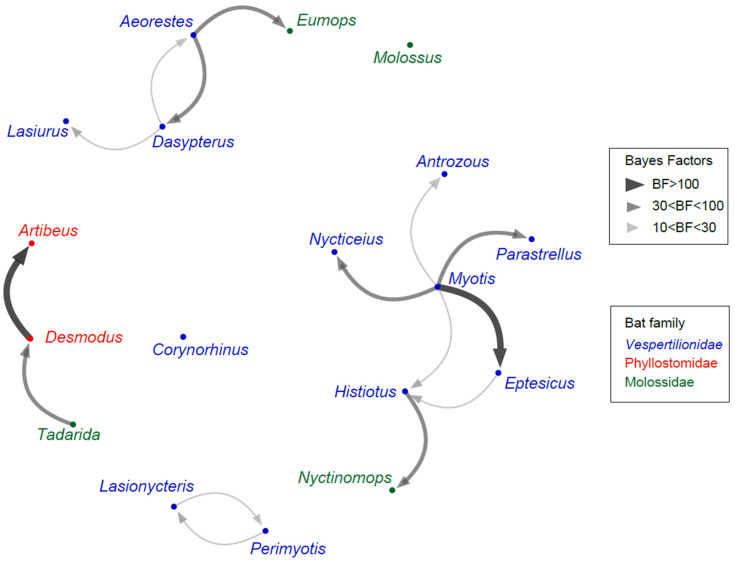
Graph representing host transition rates among bat genera. Line width indicates the switch significance level according to the Bayes factor test. Genus color indicates host family.

**Figure 4 viruses-16-01302-f004:**
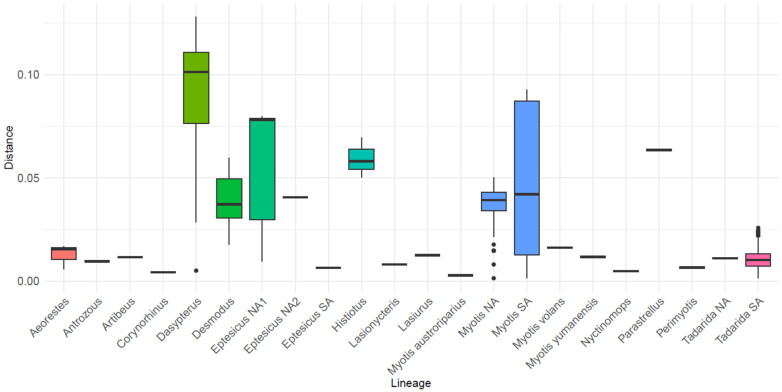
Uncorrected pairwise genetic p-distances at the lineage level. Distances were calculated by considering all sequences belonging to a specific lineage, defined as members of either a monophyletic or paraphyletic group within a given host genus. Some genera comprised multiple lineages which were treated separately.

**Figure 5 viruses-16-01302-f005:**
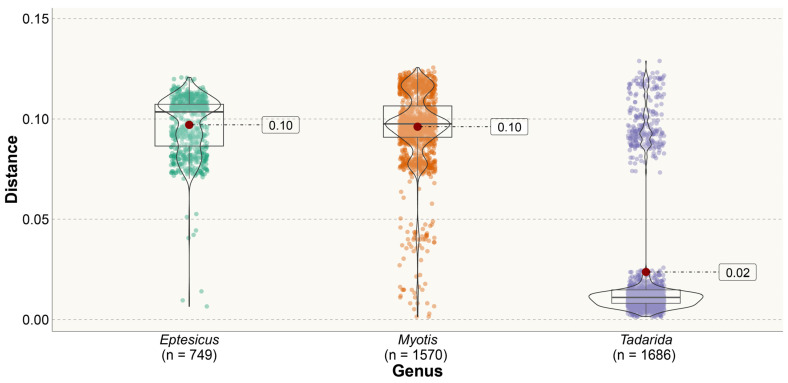
Uncorrected pairwise genetic p-distances of multilineage genera (*Eptesicus*, *Myotis*, and *Tadarida*). Violin plots and box plots show the density of data points, as well as the median (horizontal line). Dark-red dots indicate mean values.

## Data Availability

Data are contained within this article or Appendix A.

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
