# Peer review of "Bat Rabies in the Americas: Is Myotis the Main Ancestral Spreader?"

_viruses, 2024, doi:10.3390/v16081302_

Round 1

Reviewer 1 Report

Comments and Suggestions for Authors

This study examines the diversity of rabies viruses in bats across the Americas (South, Central and North America) and analyses host shifts between bat genera. The manuscript provides new insights into cross species transmission and host genera of rabies using Baysian phylogenetics based on 62 sequences generated in this study from Argentinian bats and a further 63 sequences from bats representing the main lineages. It identified a single rabies virus lineage in most bat genera that may act as a deterrent for establishment of less adapted variants and identified Myotis as the primary source of rabies transmission among bats.

The sample size is someone limited but the manuscript is well written and clear. I have only a couple of minor comments.

There is a lot of basic information in the supplementary material make it difficult to interpret at times. I would suggest at least providing some additional information in the main manuscript on the geographic locations of the bats, numbers of samples from each location and the number of samples from each genus. The conclusions then need to be drawn in the context of the number of samples within each genus/location.

The legend for Figures should be more descriptive. For example, there is no indication of what the colors mean on the tree in Figure 1.

There are a few typographical errors (e.g. CST is abbreviated multiple times throughout the manuscript).

Author Response

Comments 1: This study examines the diversity of rabies viruses in bats across the Americas (South, Central and North America) and analyses host shifts between bat generaThis study examines the diversity of rabies viruses in bats across the Americas (South, Central and North America) and analyses host shifts between bat genera. The manuscript provides new insights into cross species transmission and host genera of rabies using Baysian phylogenetics based on 62 sequences generated in this study from Argentinian bats and a further 63 sequences from bats representing the main lineages. It identified a single rabies virus lineage in most bat genera that may act as a deterrent for establishment of less adapted variants and identified Myotis as the primary source of rabies transmission among bats.

The sample size is someone limited but the manuscript is well written and clear. I have only a couple of minor comments.

There is a lot of basic information in the supplementary material make it difficult to interpret at times. I would suggest at least providing some additional information in the main manuscript on the geographic locations of the bats, numbers of samples from each location and the number of samples from each genus. The conclusions then need to be drawn in the context of the number of samples within each genus/location.

Response 1: We have added a more detailed description of the sample locations and host species, and included a new figure (Figure 1) that illustrates the distribution of the samples and sequences obtained in this study. Although we generated a significant sequence dataset, our focus extended beyond Argentinian samples to encompass a more global perspective. Although we statistically tested for geographic signals in the phylogeny, we did not perform a finer-scale analysis or an epidemiological study. In addition, our primary goal was to analyze host shifts occurring at deeper nodes in the phylogeny, rather than examining spillovers. Consequently, we included only reference sequences from major RABV hosts and did not focus on spillovers. For these reasons—namely, that we did not conduct an epidemiological survey, but instead we focused on host shifts, and because we captured the known diversity of bat RABV (as detailed in Cisterna et al. 2005)—we have not changed our conclusions regarding the number of samples from each genus or location.

Comments 2: The legend for Figures should be more descriptive. For example, there is no indication of what the colors mean on the tree in Figure 1.

Response 2: We have extended the legends to make them more descriptive.

Comments 3: There are a few typographical errors (e.g. CST is abbreviated multiple times throughout the manuscript).

Response 3: We thank the reviewer’s suggestion. We modified the manuscript accordingly.

Reviewer 2 Report

Comments and Suggestions for Authors

Understanding evolutionary processes driving viral spread is essential for disease prevention and mitigation. This study uses a Bayesian framework for inferring, visualizing, and testing hypotheses about phylogeographic history of bat-related rabies viruses (RABVs) and the factors that influence cross-species transmission (CSTs), viral spread, and diversification. This work builds on prior studies that explored these concepts but focused on a subset of RABV lineages that occur in bats within the United States (or North America). The work herein examines these questions with an expanded dataset that includes not only new sequences from Argentinian bats but also over 1800 RABV reference sequences (downloaded from Genbank) from bats throughout the Americas, thus including all major lineages. Using a trimmed dataset representing all major well-supported lineages, the authors assess the factors/parameters that influence CST within a Bayesian phylogeographic framework. Using association tests, they statistically assessed the significance of trait clustering along the branch tips and conclude that there are both host (bat genera) and geography (country) associations with viral lineages. They further use a Bayesian stochastic search variable selection approach to analyze connectivity among hosts (bat genera) and host transition (CST) rates represented in a “transmission web” to quantify host switches and assess the significance among inter-generic switches to identify which bat genera most influence CSTs (i.e., which bat hosts act as primary spreaders). From these analyses, they suggest that Myotis was the most frequent donor to other genera and may represent the ancestral “spreader” of most rabies virus diversity within the clade of bat-related RBVs.

Their phylogeny revealed that most well-supported lineages were specific to a single bat genus which they suggest supports a competitive exclusion hypothesis, wherein an adapted variant within a species may act as a deterrent for the establishment of less adapted variants. The authors further suggest that widespread bat genera (Myotis and Eptesicus) which serve as hosts for multiple unrelated RABV lineages may be reflective of geographic isolation leading to lineage extinctions and recolonizations of less-adapted variants. They do, however, suggest an alternative hypothesis for the pattern seen in Myotis. Through ancestral state reconstructions, they find that Myotis is the common ancestor to a monophyletic clade including all RABV lineages (with the exception of the Dasypterus/Lasiurus/Aorestes, plus Perimyotis and Lasionycteris) and suggest that RABV lineages evolved in Myotis through co-diversification (in the New World only) within this widespread and speciose genus. This is possible since New World Myotis spp. appear to form a monophyletic clade separate from Old World Myotis spp.

Overall, this study follows on previous studies that use sophisticated cutting-edge Bayesian statistical approaches to test key questions about the role of cross species transmission in viral emergence and spread. They found that while geographic overlap does play a role in observed global phylogenetic structure, successful CST of viruses is more likely to occur among genetically similar taxa, suggesting that genetic relatedness of hosts may reduce barriers to adaptation to a new host (i.e., most inter-generic CSTs occur among members of the same family). While this finding is not new, the expanded dataset covering all major rabies virus lineages throughout their New World distribution as well as their focus on CSTs at the level of bat genera (focusing on host shifts) show that this pattern is upheld across the entire range of bat-related RABV lineages and geographic ranges. The results of this study advance our understanding of the evolutionary and ecological influences of RABV CST and disease spread and merits publication.  A few minor comments are detailed below:

In Fig. 1 legend and lines 280-283, the authors discuss results of the ancestral host reconstruction (e.g., the pie charts in Fig. 1 represent uncertainty in ancestral host reconstruction and the Aeorestes/Dasypterus clade was proposed as the most probable common ancestor to all bat rabies lineages through ancestral host reconstruction analyses, but was qualified as not having strong support), but in the methods section, there was no detailed information on how those ancestral state reconstructions were generated. Additional detail on the methods used would be helpful.

In Fig. 1, node support associated with Bayesian posterior probabilities are indicated by node size, however, sizes are very difficult to distinguish and there is no key to the relative node sizes in the figure legend. Is there a clearer way to label nodes that would better visualize the relative BPPs?

General comment: The phylogeny generated in this study revealed that most well-supported lineages were specific to a single bat genus. This pattern is consistent with the hypothesis that genetic similarity of potential host species influences the likelihood of CST. The authors propose that another factor underlying this pattern is competitive exclusion wherein the presence of an adapted variant will outperform and prevent successful host shifting of other variants. Competitive exclusion often work on the premise that superinfection is prevented by some mechanism deployed by the initially infecting organism or that cross-immunity from prior initial infection prevents subsequent infections of related pathogens. It is not clear how these competitive exclusion mechanisms would work with RABVs given that prevalence of disease in populations is generally low and infected individuals rarely survive to become immune individuals in the population. The pattern showing monophyletic/paraphyletic clustering of most bat genera in the phylogeny can be explained by the hypothesis that genetically similar environments favor successful CST without invoking competitive exclusion. It would be helpful if the authors could expound upon the proposed mechanisms of competitive exclusion that they believe explains this pattern with RABVs.

General comment: With the general premise that CST of viruses is more likely to occur among genetically similar taxa, suggesting that genetic relatedness of hosts may reduce barriers to adaptation to a new host, there is no discussion of how CSTs from bat-related viruses to very distantly related terrestrial mammal species in the New World fits in with this paradigm. Do the authors think that these are just stochastic events; that they reflect repeated spillover events of geographically overlapping populations that in rare instances by chance are sustained in the recipient host and become evolutionary host shifts? Given that they state that understanding evolutionary processes driving viral spread is essential for disease prevention and mitigation and that terrestrial reservoirs represent a significant risk to spillover to humans and domestic animals, some discussion of these exceptions to the model would be enlightening.

Author Response

Comments 1: Understanding evolutionary processes driving viral spread is essential for disease prevention and mitigation. This study uses a Bayesian framework for inferring, visualizing, and testing hypotheses about phylogeographic history of bat-related rabies viruses (RABVs) and the factors that influence cross-species transmission (CSTs), viral spread, and diversification. This work builds on prior studies that explored these concepts but focused on a subset of RABV lineages that occur in bats within the United States (or North America). The work herein examines these questions with an expanded dataset that includes not only new sequences from Argentinian bats but also over 1800 RABV reference sequences (downloaded from Genbank) from bats throughout the Americas, thus including all major lineages. Using a trimmed dataset representing all major well-supported lineages, the authors assess the factors/parameters that influence CST within a Bayesian phylogeographic framework. Using association tests, they statistically assessed the significance of trait clustering along the branch tips and conclude that there are both host (bat genera) and geography (country) associations with viral lineages. They further use a Bayesian stochastic search variable selection approach to analyze connectivity among hosts (bat genera) and host transition (CST) rates represented in a “transmission web” to quantify host switches and assess the significance among inter-generic switches to identify which bat genera most influence CSTs (i.e., which bat hosts act as primary spreaders). From these analyses, they suggest that Myotis was the most frequent donor to other genera and may represent the ancestral “spreader” of most rabies virus diversity within the clade of bat-related RBVs.

Their phylogeny revealed that most well-supported lineages were specific to a single bat genus which they suggest supports a competitive exclusion hypothesis, wherein an adapted variant within a species may act as a deterrent for the establishment of less adapted variants. The authors further suggest that widespread bat genera (Myotis and Eptesicus) which serve as hosts for multiple unrelated RABV lineages may be reflective of geographic isolation leading to lineage extinctions and recolonizations of less-adapted variants. They do, however, suggest an alternative hypothesis for the pattern seen in Myotis. Through ancestral state reconstructions, they find that Myotis is the common ancestor to a monophyletic clade including all RABV lineages (with the exception of the Dasypterus/Lasiurus/Aorestes, plus Perimyotis and Lasionycteris) and suggest that RABV lineages evolved in Myotis through co-diversification (in the New World only) within this widespread and speciose genus. This is possible since New World Myotis spp. appear to form a monophyletic clade separate from Old World Myotis spp.

Response 1: We included this last reflection in the manuscript.

Comments 2: Overall, this study follows on previous studies that use sophisticated cutting-edge Bayesian statistical approaches to test key questions about the role of cross species transmission in viral emergence and spread. They found that while geographic overlap does play a role in observed global phylogenetic structure, successful CST of viruses is more likely to occur among genetically similar taxa, suggesting that genetic relatedness of hosts may reduce barriers to adaptation to a new host (i.e., most inter-generic CSTs occur among members of the same family). While this finding is not new, the expanded dataset covering all major rabies virus lineages throughout their New World distribution as well as their focus on CSTs at the level of bat genera (focusing on host shifts) show that this pattern is upheld across the entire range of bat-related RABV lineages and geographic ranges. The results of this study advance our understanding of the evolutionary and ecological influences of RABV CST and disease spread and merits publication.  A few minor comments are detailed below:

In Fig. 1 legend and lines 280-283, the authors discuss results of the ancestral host reconstruction (e.g., the pie charts in Fig. 1 represent uncertainty in ancestral host reconstruction and the Aeorestes/Dasypterus clade was proposed as the most probable common ancestor to all bat rabies lineages through ancestral host reconstruction analyses, but was qualified as not having strong support), but in the methods section, there was no detailed information on how those ancestral state reconstructions were generated. Additional detail on the methods used would be helpful.

Response 2: We added a brief line to clarify this aspect in the manuscript. Here, we will provide a more detailed explanation. Ancestral state reconstruction was conducted using BEAST. We first created a host genus partition with an associated trait substitution model. We chose an Asymmetric substitution model to allow for independent rate estimation between two states (A→B and B→A). Additionally, we incorporated the Bayesian Stochastic Variable Selection (BSVS) model, which aims to set certain transition probabilities to zero, thereby disallowing transitions between some states. Finally, we selected the option to “Reconstruct ancestral states at all ancestors.”

Comments 3: In Fig. 1, node support associated with Bayesian posterior probabilities are indicated by node size, however, sizes are very difficult to distinguish and there is no key to the relative node sizes in the figure legend. Is there a clearer way to label nodes that would better visualize the relative BPPs?

Response 3: We added numerical BPP values to relevant nodes in Fig. 1.

Comment 4: General comment: The phylogeny generated in this study revealed that most well-supported lineages were specific to a single bat genus. This pattern is consistent with the hypothesis that genetic similarity of potential host species influences the likelihood of CST. The authors propose that another factor underlying this pattern is competitive exclusion wherein the presence of an adapted variant will outperform and prevent successful host shifting of other variants. Competitive exclusion often work on the premise that superinfection is prevented by some mechanism deployed by the initially infecting organism or that cross-immunity from prior initial infection prevents subsequent infections of related pathogens. It is not clear how these competitive exclusion mechanisms would work with RABVs given that prevalence of disease in populations is generally low and infected individuals rarely survive to become immune individuals in the population. The pattern showing monophyletic/paraphyletic clustering of most bat genera in the phylogeny can be explained by the hypothesis that genetically similar environments favor successful CST without invoking competitive exclusion. It would be helpful if the authors could expound upon the proposed mechanisms of competitive exclusion that they believe explains this pattern with RABVs.

Response 4: We thoroughly discussed this aspect. The observed pattern, where most RABV lineages were specific to single bat genera, can indeed be explained by the hypothesis that host genetic similarity influences the likelihood of cross-species transmission (CST). Additionally, given that RABV-infected bats generally have short survival times and that prevalence in natural populations is typically low, the likelihood of co-infection with another RABV strain is minimal. We concur with the Reviewer that these factors may sufficiently explain the observed pattern without the need to invoke competitive exclusion, and have modified the manuscript accordingly.

Comment 5: General comment: With the general premise that CST of viruses is more likely to occur among genetically similar taxa, suggesting that genetic relatedness of hosts may reduce barriers to adaptation to a new host, there is no discussion of how CSTs from bat-related viruses to very distantly related terrestrial mammal species in the New World fits in with this paradigm. Do the authors think that these are just stochastic events; that they reflect repeated spillover events of geographically overlapping populations that in rare instances by chance are sustained in the recipient host and become evolutionary host shifts? Given that they state that understanding evolutionary processes driving viral spread is essential for disease prevention and mitigation and that terrestrial reservoirs represent a significant risk to spillover to humans and domestic animals, some discussion of these exceptions to the model would be enlightening.

Response 5: We have added a paragraph to the discussion addressing this important aspect. We thank the Reviewer for highlighting this issue, as it enriched our discussion and allowed us to better address the topic.